# GABA_A_ Receptor Ligands Often Interact with Binding Sites in the Transmembrane Domain and in the Extracellular Domain—Can the Promiscuity Code Be Cracked?

**DOI:** 10.3390/ijms21010334

**Published:** 2020-01-03

**Authors:** Maria Teresa Iorio, Florian Daniel Vogel, Filip Koniuszewski, Petra Scholze, Sabah Rehman, Xenia Simeone, Michael Schnürch, Marko D. Mihovilovic, Margot Ernst

**Affiliations:** 1Institute of Applied Synthetic Chemistry, TU Wien, Getreidemarkt 9/163-OC, 1060 Vienna, Austria; maria.iorio@tuwien.ac.at (M.T.I.); michael.schnuerch@tuwien.ac.at (M.S.); marko.mihovilovic@tuwien.ac.at (M.D.M.); 2Department of Molecular Neurosciences, Center for Brain Research, Medical University of Vienna, Spitalgasse 4, 1090 Vienna, Austria; florian.vogel@meduniwien.ac.at (F.D.V.); filip.koniuszewski@meduniwien.ac.at (F.K.); sabah.rehman@meduniwien.ac.at (S.R.); xenia.simeone@meduniwien.ac.at (X.S.); 3Department of Pathobiology of the Nervous System, Center for Brain Research, Medical University of Vienna, Spitalgasse 4, 1090 Vienna, Austria; petra.scholze@meduniwien.ac.at

**Keywords:** receptors, GABA_A_, benzodiazepines (BZ), binding sites, etomidate

## Abstract

Many allosteric binding sites that modulate gamma aminobutyric acid (GABA) effects have been described in heteropentameric GABA type A (GABA_A_) receptors, among them sites for benzodiazepines, pyrazoloquinolinones and etomidate. Diazepam not only binds at the high affinity extracellular “canonical” site, but also at sites in the transmembrane domain. Many ligands of the benzodiazepine binding site interact also with homologous sites in the extracellular domain, among them the pyrazoloquinolinones that exert modulation at extracellular α+/β− sites. Additional interaction of this chemotype with the sites for etomidate has also been described. We have recently described a new indole-based scaffold with pharmacophore features highly similar to pyrazoloquinolinones as a novel class of GABA_A_ receptor modulators. Contrary to what the pharmacophore overlap suggests, the ligand presented here behaves very differently from the identically substituted pyrazoloquinolinone. Structural evidence demonstrates that small changes in pharmacophore features can induce radical changes in ligand binding properties. Analysis of published data reveals that many chemotypes display a strong tendency to interact promiscuously with binding sites in the transmembrane domain and others in the extracellular domain of the same receptor. Further structural investigations of this phenomenon should enable a more targeted path to less promiscuous ligands, potentially reducing side effect liabilities.

## 1. Introduction

GABA_A_ receptors are pentameric anion channels heavily expressed in the mammalian central nervous system. They are formed of five homologous subunits, encoded by 19 different genes. Despite the significant subtype heterogeneity, it is commonly accepted that the majority of receptors is formed of two α and β subunits and a single γ subunit. The subunits form a pentameric GABA gated ion pore, with subunit interfaces formed by a principal (or +) and a complementary (or −) face of each subunit as depicted in Figure 1. Treatments of epilepsy, anxiety states and sleep disorders, as well as anesthetics, target GABA_A_ receptors [1,2]. Among these drugs, benzodiazepines are widely prescribed and are known to exert their effects via a binding site on the extracellular domain of the receptor. Specifically, the binding site of benzodiazepines is situated at the extracellular domain (ECD)- interface α+/γ− (site 1, Figure 1) and homologous with the GABA binding site at the ECD- β+/α− interface [3,4].

Benzodiazepines interact additionally with another low affinity binding site within the transmembrane domain (TMD), at the TMD-interface β+/α− (site 3) [5]. This binding site is chiefly known as the site of action for the clinically used drug etomidate, and also one of the interaction sites for propofol [6,7].

Besides the clinically relevant drugs, a wide range of compounds have been identified as modulators of the GABA_A_ receptors [1,8,9,10].

In a previous work, we identified a series of indole derivatives, sharing pharmacophore features with pyrazoloquinolinones (PQs), as GABA_A_ modulators (Figure 2a) [12]. PQs are known to bind with high affinity at the BZ site (site 1) and to modulate the receptors via the interaction with a second binding site of the ECD, at the interface between α+ and β− (site 2) [13]. An interaction of PQs with site 3 has also been reported [14] For this study, we selected the most efficacious compound of the previously published library, MTI163 (Figure 2).

The high pharmacophoric overlap of the indole derivatives in comparison with the PQs (Figure 2c,d) suggests similar binding behavior. We therefore considered the three possible binding sites as object of our studies: site 1 (ECD- α+/γ−) site 2 (ECD- α+/β−), and site 3 (TMD- β+/α−). To investigate the putative binding and mechanism of action of the new scaffold at these three sites, we performed functional and mutational studies, as well as radioligand displacement assays. Computational docking studies complement the experimental findings in the context of the recently published cryo EM structure with the PDB identifier 6HUP [4], which features diazepam in binding sites 1 and 3.

## 2. Results

### 2.1. Chemistry

The indole derivative MTI163 was synthesized as previously described [12]. DCBS192 was synthesized according to a previously reported route [13,15]. For details of the synthesis, see Appendix B.

### 2.2. Pharmacological Assessments

#### 2.2.1. Radioligand Displacement Assays

To investigate a possible binding at site 1, where PQs are binding with high affinity, we performed displacement assays with [^3^H]-flunitrazepam (Figure 3a,b). In contrast to the pyrazoloquinolinone DCBS192, which displaced the radioligand already at picomolar concentrations (Ki = 0.33 ± 0.07 nM), MTI163 did not displace [^3^H]-flunitrazepam, even at high µM concentrations (leading to a Ki of >30~µM).

Despite the similarity of the pharmacophoric features, these results exclude a binding of the indole derivative MTI163 at site 1. It seems surprising at first sight that the high pharmacophore similarity is not reflected in similar binding properties. However, we have previously observed that interactions with site 1 do not follow pharmacophore features [16], and recently published structures confirm that even ligands of the benzodiazepine chemotype do not display binding modes with aligned pharmacophore features. Specifically, flumazenil (PDB structure 6D6T [17]) and alprazolam (PDB structure 6HUO [4]) bind at the high affinity site with completely different binding modes and no overlap of pharmacophore features (Figure 3c,d).

#### 2.2.2. Functional and Mutational Studies, Beta Isoform Profile

In the next step, we aimed to compare the modulation of MTI163 in wild type receptors α1β1, α1β2, α1β3 and a concatenated version of α1β3γ2, termed α1β3γ2cct (see methods) GABA_A_ receptors (Figure 4a). MTI163 modulates GABA-elicited currents in α1β2 and α1β3 similarly. In contrast, α1β1 receptors show a much lower efficacy of MTI163 modulation (Figure 4a). In contrast to MTI163, which modulates α1β3 with very high efficacy, DCBS192 hardly modulates at all (130 ± 14% at 30 µM). The very low efficacy of this substance precluded a more detailed functional analysis and also indicates that the high similarity of pharmacophore features does not translate into similar pharmacology for these two compounds.

While modulation in the binary subunit combinations α1β2 and α1β3 does not reach saturation up to 30 µM, α1β3γ2cct and α1β1 receptors show a typical sigmoidal dose response curve (Figure 4a). The observed β isoform profile could reflect interactions with either site 2 at the extracellular domain’s α+/β− interface or site 3 at the transmembrane domain’s β+/α− interface. We thus utilized previously published mutations [18,19] to investigate their impact on modulation. β3N41R is a partial conversion of the ECD site of β3 into the β1 [18], while β2N265S is a conversion at site 3 of the β2 into β1, which is known to reverse the effects of etomidate and loreclezole [20,21].

While the mutation β3N41R did not affect the modulation of MTI163 (α1β3N41R: 1032 ± 189% vs. 1110 ± 109% in α1β3, Figure 4b), the mutation β2N265S resulted in a drop in modulation comparable to the one obtained in α1β1 (α1β2: 1108 ± 68% vs. α1β2N265S: 198 ± 26% vs. α1β1 158 ± 12%, Figure 4b).These results suggest that the extracellular α+/β− site (site 2) doesn’t contribute to the modulatory effect of MTI163, whereas the interaction with N265 located at the β+ side of site 3 in the TMD seems to be crucial for the modulation.

### 2.3. Structural Hypothesis/Computational Docking

Since all the experimental findings strongly suggest that MTI163 exerts the modulatory effect via the “etomidate site” at the TMD β+/α− site involving β2N265S, we performed computational docking to investigate the possible binding of MTI163 to the equivalent site of 6HUP which harbors diazepam. A redocking of diazepam was performed first to test whether the protocol we use recovers the experimental structure [16]. Poses with very high overlap to the one observed in the cryo-EM structure (6HUP [4]) were indeed found within the top 10 ranked docking results, evaluated with two scoring functions as well in the absence or presence of flexible sidechains (see Methods). We thus proceeded to investigate MTI163 and etomidate.

MTI163 was used for the computational investigation in its nonionized form since it was found experimentally that this acid cannot be deprotonated even under strongly basic conditions (2N NaOH). Hence, deprotonation under physiological conditions is highly unlikely. This is also supported by density functional theory (DFT) calculations, which show that an intramolecular hydrogen bond stabilizes the molecule by 11.2 kcal/mol (see Appendix B, Figure A1), providing an explanation for the low acidity observed. Additionally, our previously reported crystal structure [12] displays the molecule in the same conformation as calculated as the most stable one and hence this conformer was used for docking. For etomidate, we have no indication to select a particular conformer, it was thus docked as a fully flexible ligand.

Docking of both ligands results in a broad diversity of highly scored poses. Etomidate docking poses display high overlap with the diazepam bound state and many poses feature interactions with amino acids known to impact on the ligand’s potency and efficacy (Appendix B, Figure A2). To further disentangle etomidate’s very diverse posing space would require going to considerably higher level of theory, or to have some experimental data on its active conformation, which is out of scope for this study. The posing space accessible to MTI163 was more limited, but still featured several alternative solutions among the top 10 poses in the two scoring functions that were used here (Appendix B, Figure A3). In terms of a consensus of the top 10 poses, one featured a prominent interaction with N265, and thus it represents a highly plausible candidate for the MTI163 bound state in this site, as shown in Figure 5. Docking into the corresponding site at the β1+ interface revealed a very similarly diverse posing space (Appendix B, Figure A4).

We proceeded to examine highly ranked poses after energy minimization to obtain further insight. As expected, energy minimization improved the ligand–protein interactions. However, the minimized poses derived from consensus top 10 poses still could not be robustly ranked with standard post-docking methods at a modest level of theory. Thus, the overall conclusion from computational docking is that the diazepam site at the TMD β3+/α1− site from 6HUP can readily accommodate etomidate and MTI163, and all three ligands interact with the same set of core residues that have been shown to be relevant for the site’s modulatory effects (Figure 5b). The orientation in the pocket is very different for diazepam and MTI163, where MTI163 poses are only in a confined space parallel to the helices that form the pocket, whereas diazepam (experimental structure) and etomidate (docking results) occupy a niche that is perpendicular to the interface. While the interacting sidechains are largely the same for all three ligands, the different spatial orientation of MTI163 is not suggestive for any common binding motif that generally drives ligand recognition in this pocket.

### 2.4. Ligand Analysis

In a further step, we sought to obtain an overview concerning the frequency with which ligands have been reported to interact with multiple distinct sites on a single pentameric receptor complex, here limited to sites 1, 2 and 3, in αβγ subtypes. An exhaustive search of the literature resulted in a pool of ligands that are known for interactions with either only one, or any two, or all three sites studied here as depicted in Figure 6.

For two of these binding sites, presumably selective compounds can be found, e.g., zolpidem for the ECD site 1 and loreclezole [20,22] and tracazolate [1,23] for the TMD site 3. In contrast, for site 2 in the ECD, no exclusive ligand has been identified so far. It also needs to be kept in mind that interactions with site 2 may have gone unnoticed in the past because this site has been described in 2011 and to this date, no radioligand has been published for it.

Well-characterized showcases for promiscuous ligands are e.g., diazepam [5] and the β-Carboline DMCM [19,24], which have been described to bind at sites 1 and 3, while flurazepam binds at sites 1 and 2 (for the case of α1+β2−). The pyrazoloquinolinones CGS 9895 and LAU 177, [14] as well as the β-Carboline ZK91085 [19] were found to interact with all of the considered sites.

In addition to the cases depicted in Figure 6, several other cases of ligands using at least two distinctive sites have been reported, but the assignment of the sites is not always completely clear. Flavonoids represent a big chemical group of natural compounds and their derivatives. Many of these have been shown to be high affinity ligands of site 1 [27]; however, modulatory effects of individual compounds have been shown to occur at different sites. A prominent example is the potent 6-methoxyflavone hispidulin [28,29]. Flavanol-3-esters have been found to mimic the binding properties of loreclezole, since the positive modulatory effect of Fa131 on GABA elicited currents and was diminished by the mutations β2N265S and M236W, similarly to loreclezole/etomidate [25]. Another flavan-3-ol, Fa173, has been presented as specific antagonist of Fa131, loreclezole and etomidate. Low affinity potentiation of diazepam could also be blocked by Fa173, whereas high affinity diazepam effects, as well as modulation by neurosteroids or barbiturates were not affected—confirming the use of a common site by Fa173 and the selectively antagonized compounds [25].

Another example of a scaffold with individual ligands showing different degrees of promiscuity between the high affinity benzodiazepine site and other, unknown sites have been termed ROD compounds [30]. ROD compounds were derived from bicuculline, a GABA site antagonist, but turned out to be allosteric modulators and not bind at the GABA sites. In an effort to understand their mode of action, they were classified according to their pharmacological profile as R1 and R2 types. R1 substances bind to the benzodiazepine binding site (and completely inhibit binding of benzodiazepines) as well as another yet unknown site, named R1 site. In contrast, R2 compounds do not bind to the α+/γ− interface, nor do they inhibit benzodiazepine binding at all. The benzodiazepine effect is not additive for R1 compounds but is for R2s. Both classes were able to modulate αβ containing GABA_A_ receptors, suggesting a distinct modulatory site for both classes that differs from the benzodiazepine binding site.

Among compounds with a similar subtype profile as etomidate and loreclezole are several furanones (gamma-butyrolactones) [31]. For these, the lack of benzodiazepine site binding and sensitivity of modulatory efficacy to the β2N265S mutation were demonstrated, thus suggesting that they interact exclusively with site 3, similar to MTI163 [31].

An interesting question is whether common features determining either specificity for a particular site, or driving promiscuity, could be deduced and yield predictive structure activity rules. Given that diazepam and midazolam both interact with sites 1 and 3, while flurazepam does not interact with site 3, but interacts with sites 1 and 2, this task seems to be very complex and likely requires extensive study of focused libraries.

## 3. Discussion

In our previous work [12], we published a library of novel modulators derived from the modification of the scaffold of pyrazoloquinolinones (Figure 2). In this follow-up study, we aimed to investigate the binding sites of these novel structures. Due to the similarity of their pharmacophores, we hypothesized a binding behavior similar to the one of their parent compounds. PQs are known to bind with high affinity at the benzodiazepine binding site (site 1). More recently it was proven that they exert their modulation via a secondary binding site (site 2) [13], located between the α and β subunits in the ECD, where they bind at higher concentration. Additionally, binding at a TMD binding site (site 3) was suggested by mutational studies [14].

Here we investigated which of these binding sites are also used by MTI163 (Figure 2) and aimed to compare it directly with the identically substituted pyrazoloquinolinone DCBS192. Site 1 has the advantage to allow a direct investigation of a possible binding, employing radioligands such as tritiated flunitrazepam. Since no displacement was observed for MTI163 (Figure 3a), in contrast with the full displacement performed by DCBS192 (Figure 3b), we excluded a possible binding at site 1.

We then expanded our investigation of the pharmacological profile of MTI163, and investigated GABA_A_ receptors containing different β subunits. A marked difference in efficacy exerted by MTI163 between β2/3− and β1− containing assemblies reminiscent of etomidate’s β2/3 profile prompted mutational analysis. Conversion mutants were employed, and strongly suggest that all of the modulatory effect exerted by MTI163 originates from interactions with the binding site used by etomidate [20,26]. Taken together, all these experimental findings suggest a binding of MTI163 at site 3 only. The identically substituted pyrazoloquinolinone is a very weak modulator, and thus not accessible to functional/mutational analysis. For DCBS192 it remains thus unclear which sites it uses in addition to the high affinity binding at the benzodiazepine binding site. Thus, our experimental results indicated that the similarity in pharmacophore features between these two ligands did not at all translate into similar pharmacology.

Our efforts to generate a rational hypothesis for this finding prompted a detailed analysis of available structural and pharmacological data. Given the observation that not only pyrazoloquinolinones display activity at multiple binding sites in GABA_A_ receptors, we approached the phenomenon from the viewpoint of putative structure-activity rules (SAR) that may help to better predict for ligands their preferences for specific sites. We thus sorted known ligands into “bins” that reflect specific or promiscuous usage of the three binding sites under investigation as depicted in Figure 6. Some of the assignments of ligands to the individual bins reflect the current status of the literature, but may have to be revised in future studies, as there still is a paucity of data on the ECD α+/β– site due to lacking radioligands or antagonists. This effort revealed some interesting observations: All chemotypes that have been described so far to interact with all three sites are relatively flat, rigid and aromatic structures—the β-Carbolines and the pyrazoloquinolinones. Of note, such scaffolds also tend to display extremely high affinity for site 1, as is the case for DCBS192 with its Ki of 0.33nM. This could be the case because the benzodiazepine binding site contains a number of aromatic sidechains that facilitate pi–pi stacking of aromatic rings. A particularly impressive case of extensive pi–pi stacking at a homologous site has been reported for acetylcholine binding protein, where a sandwich of three aromatic ligands occupies the binding site (PDB structure 4BFQ, [32]). However, the reverse conclusion that planar ligands are typical for site 1 is clearly not valid, and a very wide range of scaffolds has been described for this site [1,33].

Among the best known site 1 ligands are the benzodiazepines, for which many pharmacophore models have been developed in the past—all of which rested on the assumption that most or all benzodiazepines share a common binding mode in which common pharmacophore features overlap. Based on a mutational analysis combined with computational docking, we have recently challenged this notion [16], and more recent structures in fact confirmed that benzodiazepines do not generally share a common binding mode [4,17]. Thus, there is no single consensus pharmacophore applicable to the design of site 1 ligands. Benzodiazepines also have been demonstrated to interact with sites 2 and 3, where so far only isolated studies for small numbers of ligands have been performed and for most benzodiazepines it is unknown whether they are site 1 specific or not.

Similarly, we asked whether it is possible to identify possible features that drive the interaction towards site 3. For this site, the diversity of scaffolds that was reported to elicit (sometimes β2/3− preferring) modulatory effects is also vast, Figure 6 provides several well-known examples, others are reviewed in Sieghart and Savic 2018 [1]. The only common set of features we identified was a combination of a hydrophobic feature and a hydrogen bond acceptor with a distance of from 5 to 6.5 A between these two descriptors, which is not enough to define any “core pharmacophore” for this site. Some first insights emerged from the diazepam bound 6HUP structure: In line with the large difference in affinity for the two sites (1 and 3), diazepam displays more and stronger interactions in site 1. In site 3, there are many branched hydrophobic amino acid sidechains that provide hydrophobic interactions, and can contribute with a multiplicity of rotameric states to induced fit phenomena. Moreover, this region of the receptor has been demonstrated to be highly flexible—thus, site 3 can be seen as a hydrophobic room that changes volume and shape both in response to conformational movements and in response to ligand binding events. It could be seen as a consequence that a wide variety of ligands, ranging from small molecules such as ethanol to rather large ones such as avermectin can bind with low to moderate affinity, and elicit strong modulatory effects due to the pocket position right at the hinge that couples the ECD with the TMD.

The 6HUP structure thus provides first structural insight into a phenomenon, which poses a serious challenge for rational drug development targeting GABA_A_ receptors. While the extracellular interfaces display high subtype specificity, mostly owed to the large variable segment (Loop) C and other variable parts of the ECD, the interfaces at the upper TMD where etomidate is known to bind are highly conserved—featuring only a single variable amino acid that confers β isoform selectivity. Thus, compounds that should be subtype selective (site 1 ligands) show a high propensity for unwanted low affinity effects at a more conserved site, and thus limit the subtype selectivity. In spite of the considerable size of the problem, systematic studies are lacking, and thus large scale SAR investigations cannot be performed. Due to its highly flexible nature [11], the etomidate pocket can accommodate a surprising diversity of molecules, and thus, unwanted activity there needs to be monitored carefully at early stages in ligand development until more structural data accumulates that may allow in silico screening tools into appropriate models.

Of final note, this phenomenon is not limited to the three sites discussed in the present work. Another intriguing example are bicuculline derivatives that originally were targeted at the GABA sites, many of which turned out to be modulatory ligands of the benzodiazepine site and additional unknown sites [30,34].

## 4. Materials and Methods

### 4.1. Compound Synthesis

Organic solvents were purified when necessary by standard methods or purchased from commercial suppliers [35]. Chemicals were purchased from commercial suppliers and used without further purification. Thin-layer chromatography (TLC) was performed using silica gel and 60 aluminum plates containing fluorescent indicator from Merck, and detected either with UV light at 254 nm or by charring potassium permanganate (1 g KMnO_4_, 6.6 g K_2_CO_3_, 100 mg NaOH, 100 mL H_2_O in 1M NaOH, pH = 14) with heating. Flash column chromatography (FC) was carried out on a Büchi Sepacore^TM^ MPLC system using silica gel 60 M (particle size 40–63 μm, 230–400 mesh ASTM, Macherey Nagel, Düren). Unless otherwise noted all compounds were purified with a ratio of 1/100 (weight (compound)/weight (silica)). Nuclear magnetic resonance (NMR) spectra were recorded on a Bruker *Avance Ultrashield 400* (^1^H: 400 MHz, ^13^C: 101 MHz) and Bruker *Avance IIIHD 600* spectrometer equipped with a Prodigy BBO cryo probe (^1^H: 600 MHz, ^13^C: 151MHz). The spectra are found in the supporting information (Appendix A), where chemical shifts are given in parts per million (ppm) and were calibrated with internal standards of deuterium labelled solvents CHCl_3_-*d* (^1^H 7.26 ppm, ^13^C 77.16 ppm) and deuterated dimethyl sulfoxide DMSO-*d_6_* (^1^H 2.50 ppm, ^13^C 39.52 ppm). Multiplicities are denoted by s (singlet), br s (broad singlet), d (doublet), dd (doublet of doublet), t (triplet), q (quartet), and m (multiplet). Melting points were determined with a Büchi Melting Point B-545 apparatus. High resolution mass spectrometry (HR-MS) was measured on an Agilent 6230 Liquid chromatography time-of-flight mass spectrometry (LC TOFMS) mass spectrometer equipped with an Agilent Dual AJS ESI-Source.

(E)-3-((4-bromophenyl)diazenyl)-5-chloro-1H-indole-2carboxylic acid (MTI163).

Compound MTI163 was synthesized according to a previously reported synthesis [12] in 60% yield (red solid, 30 mg). M. *p* > 300 °C; ^1^H NMR (600 MHz, DMSO-*d*6) d 7.42 (dd, *J* = 8.7, 2.2 Hz, 1H), 7.55 (d, *J* = 8.7 Hz, 1H), 7.78 (d, *J* = 8.7 Hz, 2H), 7.86 (d, *J* = 8.6 Hz, 2H), 8.43 (d, *J* = 2.1 Hz, 1H), 12.86 (s, 1H); ^13^C NMR (151 MHz, DMSO-*d*6) d 114.8 (d), 118.2 (s), 122.5 (d), 123.4 (s), 123.9 (d, 2C′), 126.1 (d),128.2 (s),132.4 (d, 2C),133.4 (s),133.9 (s, 2C), 152.0 (s), 161.6 (s); HR-MS (ESI): 377.9636 [M + H]^+^; (calcd. 377.969).

2-(4-bromophenyl)-8-chloro-2,5-dihydro-3H-pyrazolo[4,3-c]quinolin-3-one (DCBS192).

Compound DCBS192was synthesized according to reported routes in 90% yield (yellow solid, 1g). [36] M. p. decomposition > 300 °C; ^1^H NMR (400 MHz, DMSO-*d*6) d 7.60–7.66 (m, 2H), 7.71–7.76 (m, 2H), 8.16–8.18 (m, 1H), 8.18–8.24 (m, 2H), 8.8 (d, *J* = 5.5 Hz, 1H), 12.99 (s, 1H).); ^13^C NMR (151 MHz, DMSO-*d*6) d 106.1 (s), 116.1 (s), 119.9 (s), 120.3 (d), 121.2 (d), 121.8 (d), 130.4 (d), 130.8 (s), 131.6 (d), 134.3 (s), 139.2 (s), 140.0 (d), 142.4 (s), 161.6 (s); HR-MS (ESI): 373.9694 [M + H]^+^; (calcd. 373.9690).

For synthesis of precursors see Appendix B1.

### 4.2. Electrophysiology in Xenopus Laevis Oocytes

Preparation of mRNA for rat α1, β1, β2, β2N265S, β3, β3N41R, and γ2 subunits as well as α1β3γ2 concatenated constructs (α1β3γ2cct) and electrophysiological experiments with Xenopus laevis oocytes were performed as described previously [12,18,19]. α1β3γ2cct was prepared as described previously [37,38,39] and included subunits in the following order: (α1-β3-α1; γ2-β3). Oocytes were purchased from EcoCyte Bioscience (Dortmund, Germany). Cold oocytes were washed in Ca^2+^-free ND96 medium (96 mM NaCl, 2 mM KCl,1 mM MgCl2,5mM MHEPES; pH 7.5). Healthy, defolliculated cells were injected with an aqueous solution of mRNA (70 ng/µL). A total of 2.5 ng of mRNA per oocyte was injected. Subunit ratio was 1:1 for α1βx (x = 1, 2, 3, 3N41R) and 1:1 (α1-β3-α1; γ2-β3) for concatenated α1β3γ2 receptors. Injected oocytes were incubated for at least 36 h before electrophysiological recordings. Oocytes were placed on a nylon grid in a bath of NDE medium. For current measurements, oocytes were impaled with two microelectrodes, which were filled with 2 M KCl and had a resistance of 1–3 MΩ. The oocytes were constantly washed by a flow of 6 mL/min NDE that could be switched to NDE containing GABA and/or drugs. Drugs were diluted into NDE from DMSO solutions resulting in a final concentration of 0.1% DMSO. To test for modulation of GABA currents, a concentration of GABA, which elicited 3–5% in receptors of the respective maximum GABA-elicited current, was applied to the cell with increasing concentrations of compounds. All recordings were performed at room temperature at a holding potential of −60 mV using a TURBO TEC-03X npi two electrode clamp (npi electronic GmbH, Tamm, Germany). Data were digitized, recorded and measured using an Axon Digidata1550 low-noise data acquisition system (Axon Instruments, UnionCity, CA, USA). Data acquisition was done using pCLAMP v.10.5 (Molecular Devices™, Sunnyvale, CA, USA) at a 500-Hz sampling rate. Enhancement of the chloride current was defined as (I_GABA+Comp_/I_GABA_) × 100, where I_GABA+Comp_ is the current response in the presence of a given compound and GABA, and I_GABA_ is the control GABA current. The normalized current enhancement data was then used to construct dose response curves with the formula Y = Maximum/(1 + (EC_50_/x)^nH^), whereas EC_50_ is the concentration of the compound that increases the amplitude of the GABA-evoked current by 50%, and nH is the Hill coefficient. Data were analysed using GraphPadPrism v.8. for Windows (GraphPad Software, La Jolla, CA, USA, www.graphpad.com) and plotted as bar graphs or dose response curves. Data are given as mean ± SEM from at least three oocytes of two and more oocyte batches.

### 4.3. Displacement Assays

Rat cerebellar membranes were prepared and radioligand binding assays were performed as described previously [18,40]. In brief, membrane pellets were incubated for 90  min at 4 °C in a total of 500 µL of a solution containing 50  mM Tris/citrate buffer, pH  =  7.1, 150 mM NaCl and 2 nM [^3^H]-flunitrazepam in the absence or presence of either 5 µM diazepam (to determine unspecific binding) or various concentrations of DCBS192 or MTI163 (dissolved in DMSO, final DMSO-concentration 0.5%). Membranes were filtered through Whatman GF/B filters and washed twice with 4 mL of ice-cold 50 mM Tris/citrate buffer. Filters were transferred to scintillation vials and subjected to scintillation counting after the addition of 3 mL Rotiszint Eco plus liquid scintillation cocktail. Nonlinear regression analysis of the displacement curves used the equation: log(inhibitor) vs. response—variable slope with Top = 100% and Bottom = 0% Y = 100/(1  +  10^((logIC50-x) × Hillslope)^).

IC 50 values were converted to Ki values using the Cheng–Prusoff relationship [41] Ki = IC_50_/(1 + (S/KD) with S being the concentration of the radioligand (2 nM) and the KD values (4.8 nM) as described in Simeone et al., 2017 [18].

All analysis were performed using GraphPad Prism version 8.3.0 for Mac OS X (GraphPad Software, La Jolla, CA, USA, www.graphpad.com).

### 4.4. Computational Docking

Molecular docking experiments were performed with “GOLD” [42]. The recently published cryo-EM structure 6HUP was used to take advantage of the ligand bound conformation in site 3. Standard protein preparation was used, where all hydrogen atoms were added to the protein [16]. The binding site within the respective pocket was defined according to the coordinates of the diazepam - N1 of the amide group with a sphere size of 12A. Ten pocket-forming amino acids were set flexible, specifically β3/1 Thr262, Asn/Ser265 and Arg269 of the TM2 and Met286 and Phe289 of the TM3, and for α1 Gln229, Leu232 and Met236 for the TM1 and Thr265 and Leu269 for the TM2. Docking of MTI163 and etomidate was performed with all ten amino acid sidechains set flexible. Etomidate was treated as fully flexible ligand, while MTI163 was restrained to the conformer, which was observed experimentally (Appendix B, Figure A1). For the redocking of diazepam, sidechains were also either flexible or stayed fixed as they are in 6HUP, and the ligand remained fixed.

Scoring poses were primarily ranked with GoldScore and rescored with ChemScore. One-hundred GoldScore top-ranked poses were analyzed from each run. Poses that were among top 20 of the consensus pool were analyzed further. Energy minimization was performed using the MOE function “energy minimize”, selecting the Amber10:EHT force field, together with fixed hydrogens and charges, all other parameters were left at default values. Ligand interaction plots were generated, displaying the interaction of a respective docking pose of a ligand with the receptor.

## 5. Conclusions

The switch from a Pyrazoloquinoline (PQ) scaffold towards an indole based body induces striking difference in the binding properties of our substances. PQs were shown to bind at three different binding sites: (1) ECD- α+/γ−, (2) ECD- α+/β−, and (3) TMD- β+/α−, possibly also at other TMD interfaces. Surprisingly, MTI163 does not bind to binding site 1, is unlikely to exert modulation via binding site 2, but modulates GABA_A_ receptors chiefly or even exclusively through the transmembrane “etomidate” site 3. It is not obvious which properties lead to the high promiscuity of PQs or to the site 3 specificity of MTI163. Similar discrepancies in ligand-based structure activity relationships of other chemotypes have been observed previously, but have not yet been systematically studied. Several benzodiazepines also interact with site 3, flurazepam interacts with site 2, while alprazolam and flumazenil seem to be more specific to site 1 only. This particular type of ligand promiscuity is very likely to be missed in standard assays aimed at refining interactions with a single site. With this study, we hope to raise awareness and to prompt further studies dedicated to this theme.

## Figures and Tables

**Figure 1 ijms-21-00334-f001:**
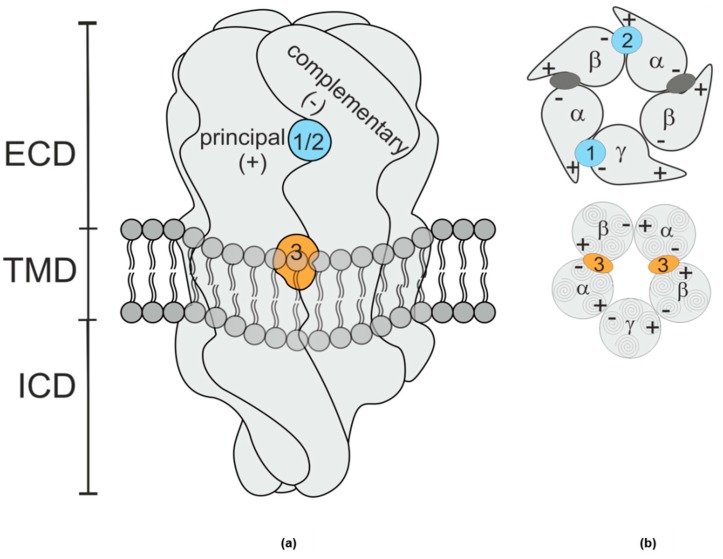
(**a**) Schematic rendering of a GABA_A_ receptor. The extracellular domain (ECD) and transmembrane domain (TMD) contain various binding sites [11]. The binding sites studied in this work are at interfaces formed by a principal (+) subunit face and a complementary (−) subunit face. The shape resembles vaguely a space filling rendering of the protein’s ECD and TMD, while intracellular domain (ICD) shape is a purely schematic rendering based on more remote homologues. (**b**) Schematic planes through the ECD (top) and TMD (bottom) of a canonical αβγ receptor, the GABA sites are indicated as dark grey ellipses. Site 1 (blue), the high affinity benzodiazepine site, is at the ECD– α+/γ− interface; site 2 (blue), which confers modulatory effects of pyrazoloquinolinone ligands is at the ECD– α+/β− interface, and site 3 (orange), the etomidate site [7] occurs twice and is located at the TMD– β+/α− interfaces below the GABA binding sites.

**Figure 2 ijms-21-00334-f002:**
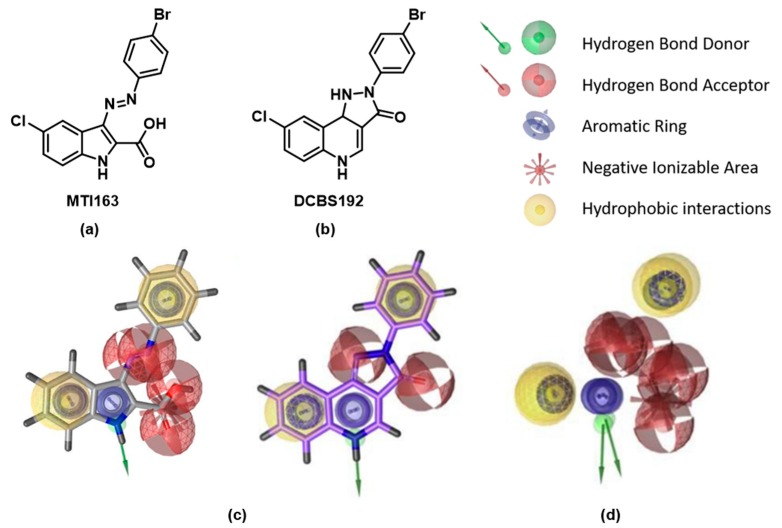
(**a**) Structure of MTI163; (**b**) structure of DCBS192; (**c**) pharmacophores of indole derivatives (left) and of PQs (right); (**d**) superposition of pharmacophores of PQ and indole derivatives. Panels c and d depict the pharmacophores already discussed in [12].

**Figure 3 ijms-21-00334-f003:**
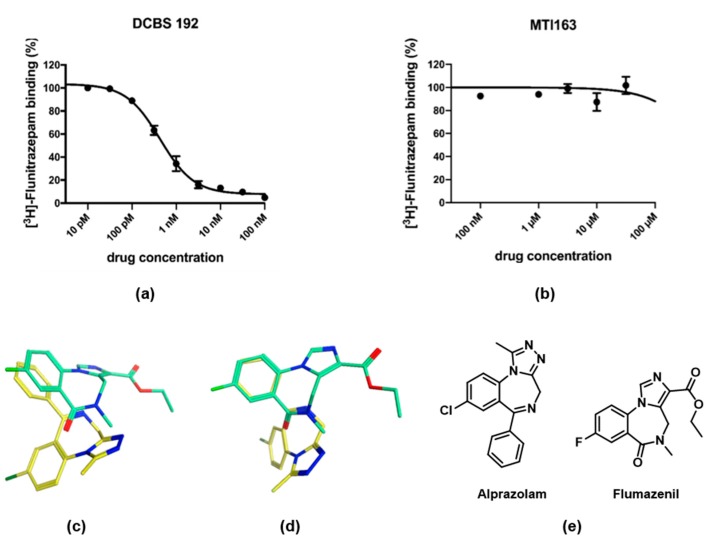
Radioligand displacement assay of [^3^H]-Flunitrazepam by DCBS192 (**a**) and MTI163 (**b**). Membranes from rat cerebellum were incubated with 2 nM [^3^H]-flunitrazepam in the presence of various concentrations of the two ligands. Data shown are the mean ± SEM of three independent experiments performed in duplicates each. (**c**,**d**) Superposition of the flumazenil bound 6D6T (green) and the alprazolam bound 6HUO (yellow) in two different perspectives demonstrate that the structurally related molecules do not bind with aligned pharmacophore features at all. The heteroatoms of the molecules are depicted in different colors: blue-nitrogen, red-oxygen, green-chloro. (**e**) Structures of alprazolam and flumazenil.

**Figure 4 ijms-21-00334-f004:**
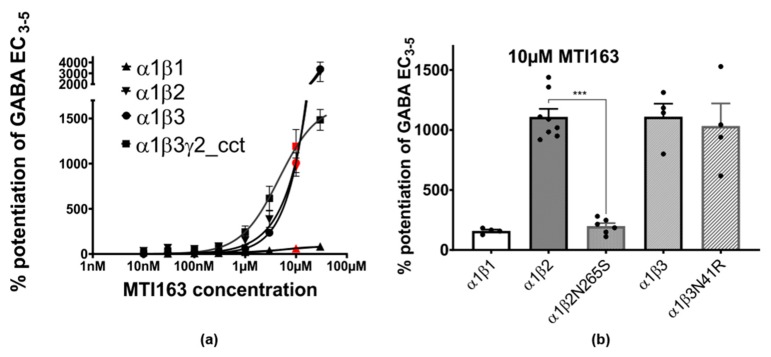
(**a**) Concentration-dependent MTI163 modulation of GABA EC_3-5_ elicited current in α1β1, α1β2, α1β3 and α1β3γ2cct. Response at each point of measurement is represented as mean ± SEM (*n* = 3–4). Response at 10 µM is highlighted in red color. The *y*-axis is broken at indicated values for better visualization of all curves in the panel. For better visualization, a full scale version of the α1β1 dose response curve is in the Appendix B (Figure A5). (**b**) Comparison of MTI163 modulation of GABA_3-5_ elicited current in α1β1, α1β2, α1β2N265S, α1β3, and α1β3N41R receptors at 10 µM. Bars are given as mean ± SEM (*n* = 4–8). Each individual data point is displayed by a dot. Statistically significant differences were determined by one-way ANOVA with Tukey’s multiple comparisons post-hoc test, *** corresponds to *p* < 0.0001.

**Figure 5 ijms-21-00334-f005:**
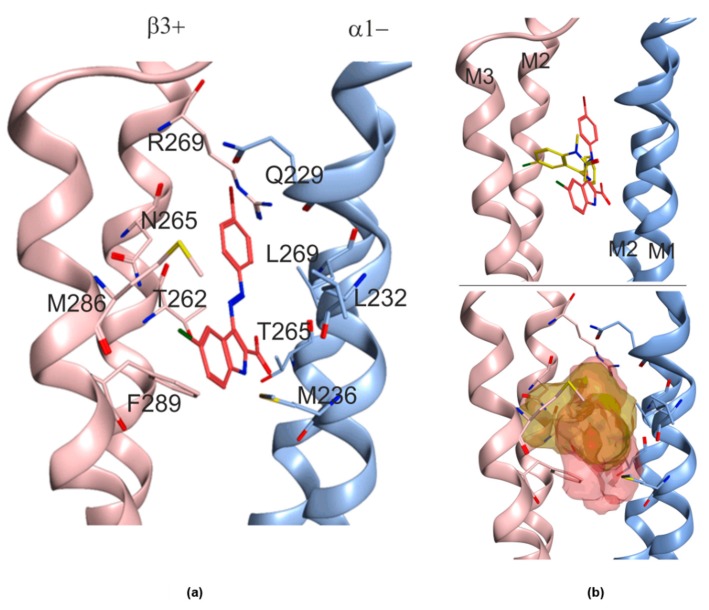
Representative docking pose of MTI163 in the diazepam pocket of 6HUP. (**a**) The MTI163 pose, which is fully consistent with all experimental evidence, is displayed with the pocket-forming amino acids that were set flexible during docking. (**b**) Overlay of diazepam (yellow) and the MTI163 (red) pose in 6HUP ligands structures (top), ligands surfaces (bottom). The left subunit (pink) is β3, the right one (blue) is α1.

**Figure 6 ijms-21-00334-f006:**
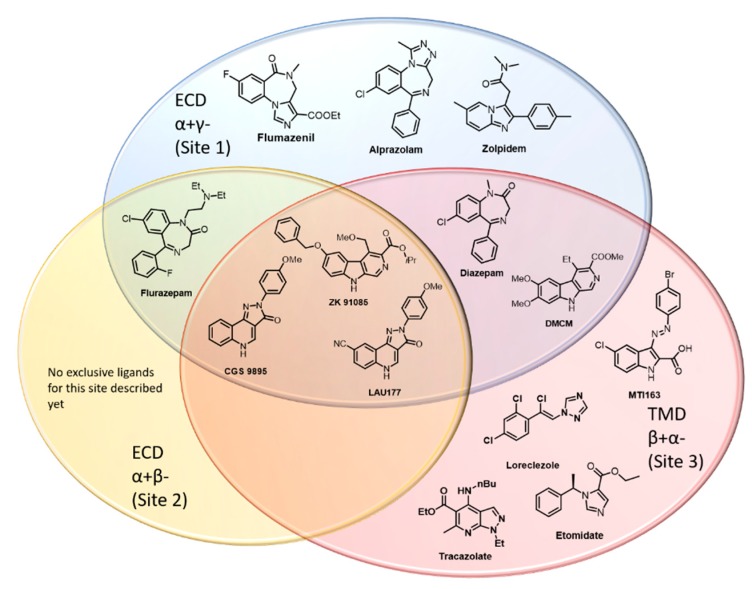
Graphical overview of chemical scaffolds which bind to distinctive interfaces at αβγ GABA_A_ receptors. The three binding sites are depicted as colored ellipses: site 1, ECD- α+/γ−: blue; site 2, ECD- α+/β−: orange; site 3, TMD- β+/α− red. Substances that bind at more than one site are placed at the intersection of the different binding sites. All placements rest on current evidence from structural studies [4,17] and decades of mutational studies [19,20,21,23,24,25,26]. Some of the placements are thus tentative. Of note, for both the benzodiazepine site and the “etomidate site”, many more ligands have been described, partly reviewed by Sieghart (2015) and Sieghart and Savic (2018) [1,10]; however, low affinity interactions with other sites have not been regularly investigated.

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
