# Peer review of "GABAA Receptor Ligands Often Interact with Binding Sites in the Transmembrane Domain and in the Extracellular Domain—Can the Promiscuity Code Be Cracked?"

_ijms, 2020, doi:10.3390/ijms21010334_

Round 1

Reviewer 1 Report

This manuscript describes a pyrazoloquinolone derivative that at first glance, was proposed to have structure-activity similar to its derived class at the alpha-beta interface, however surprisingly demonstrated a novel GABAA receptor pharmacophore with disparate ligand binding properties. The authors are thorough in their attempt to classify the features of the interaction with MTI163, and they create a compelling argument for further in-depth investigation of binding properties beyond what a drug structure-activity may appear to be. The experiments are well-thought out and the evidence strongly supports their findings.

Major concerns:

There are two Figure 1's, the GABAA receptor cross section figure and the molecular structures for MTI163 and DCBS192. This causes the other figures to be mislabeled, and makes the data difficult to follow.

The Figure with the "graphical overview of chemical scaffolds..." (venn diagram) should be labeled with "site 1", "site 2", and "site 3", in addition to the ECD or TMD nomenclature, since the authors consistently reference both in the text. This would greatly aid the reader in referring between the figure and the text.

Minor concerns:

Lines 110 (page 5) and 131-133 (page 6): The numbers in this paragraph should have some units, or some indication of the measurement being presented in the text without the reader having to refer back to the Figure. Still, the Figure from which the data is taken should also be cited within this paragraph.

Line 175 (page 7) robustly is misspelled as "rubustly"

Line 222-230 (page 9) It may help to illustrate the point at hand by explicitly explaining the ROD compounds were derived from bicuculline but had different structure-activity and binding properties (this is mentioned in the discussion, but would be suited to be put earlier, here)

Line 331 (page 11) The abbreviation TLC should be explained, as there is no prior context for what it is.

Line 415 (page 13) The "MOE function" used is not explained in any detail that would be helpful to the reader in order to replicate the study

Line 426 (page 13) "Similar cliffs in structure..." It is not clear what the author means by the word cliffs in this context, and it does not seem technical nor descriptive.

Author Response

Reviewer 1

Comments and Suggestions for Authors

This manuscript describes a pyrazoloquinolone derivative that at first glance, was proposed to have structure-activity similar to its derived class at the alpha-beta interface, however surprisingly demonstrated a novel GABAA receptor pharmacophore with disparate ligand binding properties. The authors are thorough in their attempt to classify the features of the interaction with MTI163, and they create a compelling argument for further in-depth investigation of binding properties beyond what a drug structure-activity may appear to be. The experiments are well-thought out and the evidence strongly supports their findings.

Major concerns:

There are two Figure 1's, the GABAA receptor cross section figure and the molecular structures for MTI163 and DCBS192. This causes the other figures to be mislabeled, and makes the data difficult to follow.

In the version we submitted as well as the one which the journal provided for revision, the Figure numbering is correct – we assume that the pdf export may have duplicated a figure.

The Figure with the "graphical overview of chemical scaffolds..." (venn diagram) should be labeled with "site 1", "site 2", and "site 3", in addition to the ECD or TMD nomenclature, since the authors consistently reference both in the text. This would greatly aid the reader in referring between the figure and the text.

These labels were added.

Minor concerns:

Lines 110 (page 5) and 131-133 (page 6): The numbers in this paragraph should have some units, or some indication of the measurement being presented in the text without the reader having to refer back to the Figure. Still, the Figure from which the data is taken should also be cited within this paragraph.

We corrected this in all instances where we only provided numbers to now read e.g. 158% ±12% and cited the figure.

Line 110, that in the file I have is page 4, has the reference to the figure. I am not sure what they meant with the units. Maybe we can write % of potentiation? Next to the numeric values?

Line 175 (page 7) robustly is misspelled as "rubustly"

The typo was corrected

Line 222-230 (page 9) It may help to illustrate the point at hand by explicitly explaining the ROD compounds were derived from bicuculline but had different structure-activity and binding properties (this is mentioned in the discussion, but would be suited to be put earlier, here)

We added the sentence “ROD compounds were derived from bicuculline, a GABA site antagonist, but turned out to be allosteric modulators and not bind at the GABA sites.” We agree that this makes the example more accessible.

Line 331 (page 11) The abbreviation TLC should be explained, as there is no prior context for what it is.

Done. And added to list of abbreviation

Line 415 (page 13) The "MOE function" used is not explained in any detail that would be helpful to the reader in order to replicate the study

We expanded to “Energy minimization was performed using the MOE function “energy minimize”, selecting the Amber10:EHT force field, together with fixed hydrogens and charges, all other parameters were left at default values.”

Line 426 (page 13) "Similar cliffs in structure..." It is not clear what the author means by the word cliffs in this context, and it does not seem technical nor descriptive.

We reworded the sentence which now reads “Similar discrepancies in ligand-based structure activity relationships of other chemotypes have been observed previously… “. Cliffs in structure activity relationship landscapes is “field specific jargon”, and we thank the reviewer for pointing out the need to reword.

Reviewer 2 Report

1) L37, Any reason that authors mention GABA receptor in postnatal brain because it most likely as "excitatory” GABA. 

2) delete Figure in page. duplcated with page 3.

3) page 3, Figure 1 similar to Figure 1 in author's another paper in Bioorganic & Med Chem, 2019, 27, 3167.

4) page 4, Figure 2 similar to Author's Figure 2 in Eur J Med Chem, 2019, 180, 340e349.

4) All Figure numbers need changed, e.g. Figure 1 in page 4 actually is Figure 2....... sometimes cause confusing in results and discussion sections.

5) L65, Is figure 1 Figure 2?

6) Page 5, Figure 3, any flumazenil (6D6T [16]) or alprazolam binding curves as shown in a and b? Why shown alprazolam and Flumazenil in Figure 4 a-e?

7) L110, sample number?

8) Labels for MYI163 and diazepam in Figure 5 (page 7)

9) page 8, add site 1 to 3 in Figure 6 also.

10) L220, why indicate neurosteroids and barbiturates here? Not mentioned in any other places, even not in Figure 1 and paragraph (add a little bit more information regarding neurosteroids and barbiturates in the introduction section or figure 1?).

11) L333, how about pH value.

12) How about basal GABA currents in Xenopus laevis oocytes before transfection in 4.2 of page 12. 

13) page 12, Axon Ins is Molecular Devices NOW. What is sample rates? Did you normalized current as pA/pF before comparing between groups if applicable?

Author Response

Reviewer 2

Comments and Suggestions for Authors

1) L37, Any reason that authors mention GABA receptor in postnatal brain because it most likely as "excitatory” GABA.

We are not sure what the reviewer requests – we wanted to clearly indicate that the prenatal brain is different with respect of inhibitory/ excitatory effects of GABA.  Since it is not essential for the clinical relevance, we deleted the half sentence.

2) delete Figure in page. duplcated with page 3.

It seems that the pdf export from our submitted work contained a duplicated figures – in our submission, and also in the revised MS, there are no duplicated figures.

3) page 3, Figure 1 similar to Figure 1 in author's another paper in Bioorganic & Med Chem, 2019, 27, 3167.

The cartoon image is indeed similar, as it depicts the same generic pentamer. The specific illustration was made for this paper, with the binding sites of interest depicted. Since the differences can clearly be seen, we assume that this is not a case of duplication.

4) page 4, Figure 2 similar to Author's Figure 2 in Eur J Med Chem, 2019, 180, 340e349.

The figure was made for this paper. To depict these pharmacophores with Ligandscout results in near-identical looking individual panels – but the way the panels are arranged is different, and this figure also displays the ligands used in this study. Thus, we also consider this a unique figure, depicting the same scientific contents that is needed in both papers to follow the line of reasoning. We added to the legend a sentences stating “Panels c and d depict the pharmacophores already discussed in [12].”.

4) All Figure numbers need changed, e.g. Figure 1 in page 4 actually is Figure 2....... sometimes cause confusing in results and discussion sections.

We verified that in our original submission, and also in the revised MS, there are no duplicated figures and that all figures are numbered consistently. We suspect that the pdf export from what we submitted did not correctly reproduce the MS.

5) L65, Is figure 1 Figure 2?

In the version I am looking at, we refer in line 66 to figure 2, I am not sure why the reviewer  sees fig 1, see also comments above.

6) Page 5, Figure 3, any flumazenil (6D6T [16]) or alprazolam binding curves as shown in a and b? Why shown alprazolam and Flumazenil in Figure 4 a-e?

The composite image conveys two messages: The experimental results in panels a and b show the lack of affinity of MTI163 for site 1. The other panels document that benzodiazepines do not bind in site 1 according to pharmacophore alignments. We have expanded the text a bit to make the transition from our experimental data to published structural evidence against pharmacophore based interactions clear, and we hope that it is now less confusing.

7) L110, sample number?

The line numbers the reviewer saw are not totally consistent with what we submitted. We fail to understand what the reviewer refers to, and cannot provide a correction or explanation.

8) Labels for MYI163 and diazepam in Figure 5 (page 7)

The ligands are now identified in the legend by the colors used in the rendering (diazepam in yellow and MTI163 in red).

9) page 8, add site 1 to 3 in Figure 6 also.

We have added the site number labels to the figure.

10) L220, why indicate neurosteroids and barbiturates here? Not mentioned in any other places, even not in Figure 1 and paragraph (add a little bit more information regarding neurosteroids and barbiturates in the introduction section or figure 1?).

We discussed the ability of Fa173 to antagonize some (etomidate) and not other (steroid, barbiturate) modulatory ligands. Otherwise the barbiturates and neurosteroids are not topic of our work. We hope that rewording this passage makes the context clear now.

11) L333, how about pH value.

Is corrected, pH is reported

12) How about basal GABA currents in Xenopus laevis oocytes before transfection in 4.2 of page 12. 

It is well established that sham injected Xenopus laevis oocytes do not display GABA induced currents.  In similar papers no such experiments are reported.

13) page 12, Axon Ins is Molecular Devices NOW. What is sample rates? Did you normalized current as pA/pF before comparing between groups if applicable?

We have added to the methods section sampling rate and the normalization. We provide our materials “as bought”, i.e. equipment that was bought longer ago from Axon Instruments is listed as such, while more recently purchased items correctly give Molecular Devices as provider.